# Towards Personalized AI: Early-stopping Low-Rank Adaptation of Foundation Models

## Abstract

Foundation models, such as Latent Diffusion Models and Generative Pre-trained Transformers, trained on broad data have shown impressive results in various downstream applications. Fine-tuning a pre-trained foundation model is an affordable way to customize it on small and personalized data. However, the non-AI experts often struggle with the hyperparameter configurations and sometimes encounter the overfitting issue without even realizing it. To mitigate this issue, we introduce a new monitoring metric (CS-Fluctuation) to facilitate early stopping the fine-tuning process. Specifically, we leverage Low-Rank Adaptation (LoRA) to fit the small scale of the personalized data while monitoring the cosine similarity of the parameter changes between the LoRA branch and its corresponding layer. When the changes become steady, we observe the onset of overfitting issue which becomes increasingly severe as fine-tuning progresses. Empirically, we leverage various types of personalized data to conduct customization experiments on both vision and language foundation models, which corroborates the effectiveness of CS-Fluctuation in early stopping the LoRA fine-tuning. The code can be found at the anonymous link: `https://anonymous.4open.science/r/EarlyStopLoRA-7467/`.

## 1 Introduction

Foundation models Bommasani et al. (2021) that are trained on broad data have demonstrated impressive results in various downstream applications. For example, the Generative Pre-trained Transformers (GPTs) (Brown et al., 2020) are trained from a vast amount of text data, which fostered a powerful ChatGPT OpenAI (2023) for conversational applications. Latent Diffusion Models (LDMs) (Rombach et al., 2022), whose encoder and decoder are pre-trained from large-amount of images, are customized into photorealistic text-to-image generation (CompVis, 2022), image editing (Zhang & Agrawala, 2023), etc.

For personalized AI, fine-tuning a pre-trained foundation model is an affordable way to take advantage of its broad capabilities using a set of small and personalized data. Ruiz et al. (2023) proposed Dreambooth which finetunes the whole LDM using a few images and then synthesizes photorealistic images of different scenes. A more efficient finetuning technique is Low Rank Adaption (LoRA) (Hu et al., 2021). LoRA performs the low-rank decomposition of the transformer structure, which significantly reduces the cost of fine-tuning the large foundation models (Dettmers et al., 2023; Cuenca & Paul, 2023). Thus, LoRA enables small companies or individuals to customize a foundation model on a small dataset and even fine-tune the private datasets using their local machines.

However, LoRA can easily overfit a small set of training data, causing a barrier to the customization of foundation models by non-AI experts. As shown in Figure 1, an individual wants to incorporate the personalized photos or texts into a foundation model, such as a pretrained LDM or LLM, but can only provide limited references. As the finetuning progresses, LoRA can quickly learn the small reference data, leading to overfitting. As shown in Figure 1a, the overfitted LoRA even generated the original reference images. Even worse, individuals often lack sufficient validation data to early stop the fine-tuning process. In many cases, such as personalized image generation, reliable evaluation metrics are also be absent. The above issues require a validation-independent criterion to early stop the fine-tuning process.

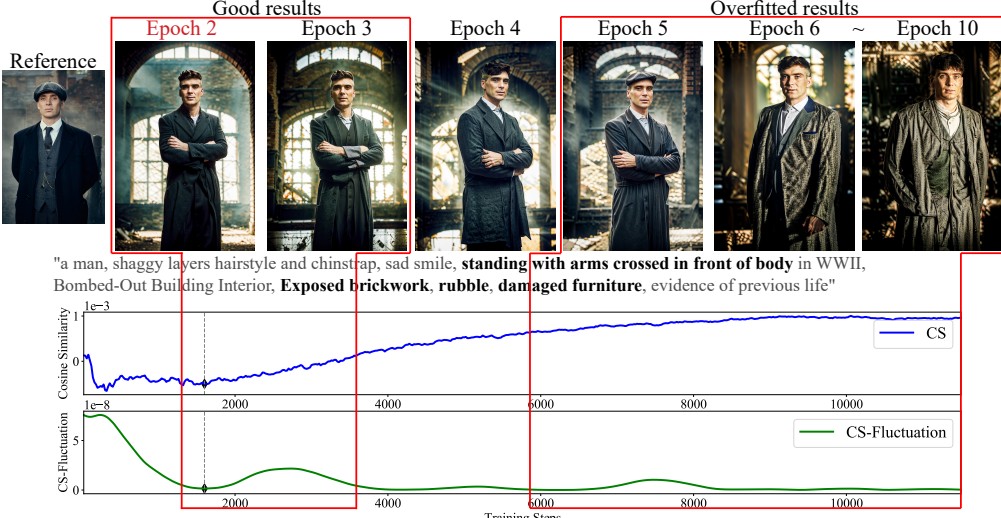

(a) Use LoRA to Fine-tune a LDM to generate the personalized images according to the reference data. In this case, an individual expects vivid photos that are consistent with the provided prompt and containing his/her face. Generated images should preserve high quality and manifest diversity in aspects such as hairstyles, scenes, and attire. Besides, the generated images do not exhibit distortions or overly resemble the provided reference data. As illustrated in the figure, the overfitted LoRA models ignore the provided prompt such as "arms crossed", and impact image quality, leading to blurring and loss of details.

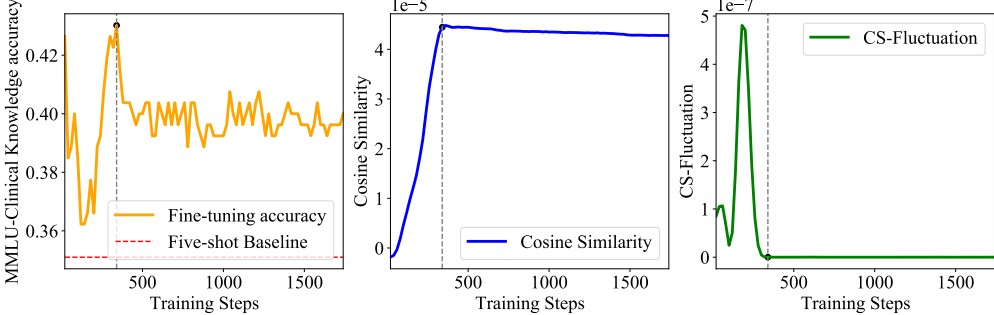

(b) Use LoRA to Fine-tune a LLM on a small and personalized texts. In this case, an individual provides a small amount of texts related to "clinical knowledge" and expects that the fine-tuned LLM with LoRA can has good generalization performance on unseen "clinical knowledge" texts. The Five-shot baseline accuracy comes from Touvron et al. (2023a).

Figure 1: LoRA of a LDM and LLM. As the LoRA fine-tuning progresses, the cosine similarity (CS) between LoRA layer and its corresponding original layers undergoes abrupt changes before settling into a more gradual and stable pattern. Our proposed monitoring metric—CS-Fluctuation—monoter the fluctuations of the CS changes. When CS-Fluctuation becomes small, it strongly suggests early stopping the fine-tuning process. The grey dashed line is the turning point that is located by our proposed algorithm, e.g., the Epoch 2 in red in Fig.1(a) is where the turning point is located.

To early stop the fine-tuning process, we introduce a new monitoring metric, called CS-Fluctuation. Initially, we observe the fine-tuning process of cosine similarity (CS) of parameters between the low-rank layer (used in the LoRA branch) and its original layer (used in a pretrained foundation model). During the LoRA fine-tuning for both LDM and LLM, we observe that CS undergoes abrupt changes before settling into a more gradual and stable pattern, despite the different learning rate chosen. Interestingly, the turning point of overfitting happens exactly at the transitional point of CS from "abrupt" changes to "stable" changes, as shown by blue lines in Figure 1. Thus, to aid in locating the turning point (grey dashed lines), we propose a new metric called CS-Fluctuation (denoted as green lines), where we calculate the variance of CS slopes across training iterations and apply moving average techniques (Box et al., 2015) to smoothen the curve.

To verify the effectiveness of CS-Fluctuation in fine-tuning foundation models utilizing LoRA technology, we chose several high-quality LDMs from CivitAI (Civitai, 2022) and open-source LLaMA series (Touvron et al., 2023a) of LLMs as base models. A series of fine-tuning experiments are conducted on multiple small-scale image and text datasets, aiming to simulate the real situations of personalizing foundation models. The experimental results revealed that the CS-Fluctuation can effectively identify the turning point to early stop the process of fine-tuning, thus avoiding the issues of overfitting. In practice, the LoRA models corresponding to these turning points demonstrated better performance in most cases. Adopting such a strategy of early stopping the fine-tuning process before the onset of overfitting reduces unnecessary consumption of computational resources.

## 2 RELATED WORK

### 2.1 FOUNDATION MODELS

Foundation models (Bommasani et al., 2021) such as BERT (Devlin et al., 2018), GPT-3 (Brown et al., 2020), CLIP (Radford et al., 2021), have demonstrated superior performance in solving various types of complex tasks. This paper focuses on two types of foundation models, i.e., text-to-image latent diffusion models (LDMs) and large language models (LLMs), which are reviewed as follows.

**Diffusion models** (DMs) have displayed remarkable performance in image synthesis. Compared with other generative models such as GANs (Goodfellow et al., 2014; Brock et al., 2018), DMs can mitigate the issues of training instability and mode collapse Ho et al. (2020); Song et al. (2020). Moreover, DMs can model highly complex distributions of natural images without requiring large amount of parameters Razavi et al. (2019).

Notably, text-to-image diffusion models have attracted extensive attention. To generate photorealistic images, the GLIDE (Nichol et al., 2021) introduced text conditions during the diffusion process, while the DALL-E2 (Ramesh et al., 2022) enhanced the precision of text and image alignment through the integration of the CLIP (Radford et al., 2021) joint feature space. Notably, Latent Diffusion Models (LDMs) Rombach et al. (2022) perform the denoising processes in the latent space, which can effectively reduce computational resources while maintaining the quality and flexibility of generated images. LDMs have facilitated the emergence of popular image editing tools like ControlNet (Zhang & Agrawala, 2023), Instruct-Pix2Pix (Brooks et al., 2023) and Adetailer (Bingsu, 2023), benefiting artists and designers.

**Language Models** (LMs) have demonstrated their potentials in solving complex tasks across various domains (Touvron et al., 2023b). LMs adopt the transformer architecture (Devlin et al., 2018) and the attention mechanism (Vaswani et al., 2017) and various pre-training techniques. Then, a pretrained LM can be fine-tuned for specific applications. Recently, Brown et al. (2020) has shown that the large language Models (LLMs) have outstanding few-shot learning capabilities and can adapt downstream tasks efficiently. The LLM examples are GPT series (Radford et al., 2018; 2019; Brown et al., 2020; Floridi & Chiriatti, 2020; OpenAI, 2023) and LLaMA series (Touvron et al., 2023a;b), which can demonstrate human-level performance.

### 2.2 PERSONALIZED AI

AI has recently shifted from universal models to personalized solutions, emphasizing that AI should meet individual needs rather than providing a "one-size-fits-all" approach. The rapid development of large foundation models has enabled the lightweight personalized AI, which allows users to obtain a high-performing AI model with just a few reference data. For example, Gal et al. (2022) proposed a personalized text-to-image generation, which can synthesize novel scenes of user-provided reference images. Ruiz et al. (2023) proposed Dreambooth that allows personalized and diversified scene renderings. Besides, Kumari et al. (2023) proposed Custom Diffusion to synthesize userprovided reference concepts. However, those methods require smart framework designs and careful hyperparameter configurations, which can be barriers for non-AI experts.

In contrast, **Low-Rank Adaptation** (LoRA) provides a unified solution for fitting a small amount of personalized data. The LoRA technique (Hu et al., 2021) was initially developed for the efficient fine-tuning of LLMs and has also been extended to LDMs (Cuenca & Paul, 2023). LoRA freezes the pre-trained model weights and introduces the trainable low-rank counterparts, which can

greatly reduce the number of trainable parameters. Consequently, LoRA lessens the demand for computational resources and offers versatile customization tailored to small and personalized data.

## 3 METHOD

In this section, Section 3.1 reviews the preliminaries of LoRA. Section 3.2 introduces our proposed monitoring metric, i.e., CS-Fluctuation as well as an algorithm of early stopping the fine-tuning process.

### 3.1 PRELIMINARIES OF LORA

LoRA (Hu et al., 2021) and QLoRA (Dettmers et al., 2023) are efficient fine-tuning techniques that were designed for LLMs. LoRA can be also utilized in other foundation models such as LDM (Cuenca & Paul, 2023). LoRA is a low-rank decomposition of foundation models, which can significantly reduce the number of trainable parameters and memory usage during the fine-tuning process of downstream tasks. Furthermore, a pre-trained foundation can be used to build many small LoRA branches for different tasks. The details are elaborated in Figure 2 and Equation 1.

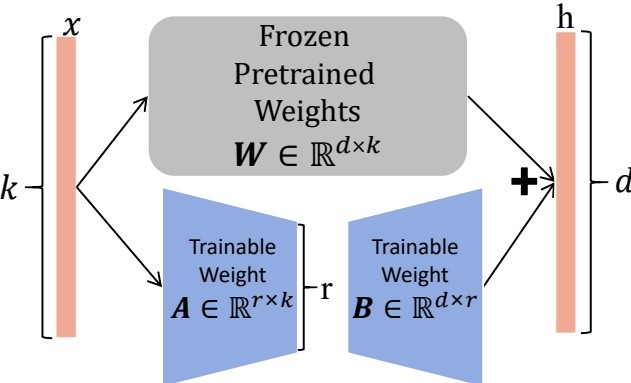

Figure 2: In the LoRA branch, the matrices $A$ and $B$ are trainable and can efficiently fit the small amount of personalized data

$$\text{h} = \boldsymbol{W}x + \boldsymbol{B}\boldsymbol{A}x \tag{1}$$

When we use LoRA to fine-tune on a small personalized data, the original pre-trained weight matrix $\boldsymbol{W} \in \mathbb{R}^{d \times k}$ is frozen. However, we concatenate a trainable LoRA branch, in which the shape of the matrix $\boldsymbol{B}\boldsymbol{A}$ equals that of $\boldsymbol{W}$, where $\boldsymbol{B} \in \mathbb{R}^{d \times r}$, $\boldsymbol{A} \in \mathbb{R}^{r \times k}$, and the rank $r$ is significantly less than $d$ or $k$. To fine-tune the foundation models, LoRA commonly operates on transformer architecture. Specifically, LoRA targets the each layer in the attention block.

### 3.2 CS-FLUCTUATION: TRACKING LEARNING STATUS TO AVOID OVERFITTING

In this section, we propose a CS-Fluctuation metric to monitor the LoRA fine-tuning process. Once CS-Fluctuation becomes steady and small, we early stop the fine-tuning process to avoid overfitting.

CS-Fluctuation is computed based on the Cosine Similarity (CS) between frozen parameters in a pre-trained foundation model ($\boldsymbol{W}$) and their counterparts in LoRA ($\boldsymbol{B}\boldsymbol{A}$). CS is defined in Equation 2 as follows.

$$\text{CS}(\boldsymbol{B}\boldsymbol{A}, \boldsymbol{W}) = \frac{1}{N} \sum_{i=1}^{N} \frac{\text{vec}(\boldsymbol{B}_i \boldsymbol{A}_i) \cdot \text{vec}(\boldsymbol{W}_i)}{\|\text{vec}(\boldsymbol{B}_i \boldsymbol{A}_i)\| \|\text{vec}(\boldsymbol{W}_i)\|}, \tag{2}$$

where $\text{vec}(\cdot)$ denote the vectorization that flattens the matrix to one-dimensional vector, and $i$ is the index of layers, in which there is a LoRA counterpart to perform a low-rank decomposition.

To calculate CS-Fluctuation, we apply the technique of moving window average $MA(\cdot)$ on CS and it slope $\nabla$CS in the batch-wise matter.

$$\mathrm{MA(CS}_j) = \frac{1}{M} \sum_{j}^{j+M} \mathrm{CS}_j, \tag{3}$$

where $j$ is the index of iteration steps in fine-tuning process, and $M$ is the size of moving window. Note that in order to calculate the CS of the iteration $j$, we need to calculate $M$ more iterations of fine-tuning.

Now, we calculate the smoothed version of CS slope, i.e, $X_j = \mathrm{MA}(\nabla(\mathrm{MA(CS}_j))$. For simplicity, we approximate the CS slope by $\nabla\mathrm{CS}_j = \mathrm{CS}_j - \mathrm{CS}_{j-1}$. The CS-Fluctuation is then the variance value of smoothed CS slope, as shown in Equation 4.

$$\mathrm{CS\text{-}Fluctuation(CS}_j) = \frac{1}{\mathrm{lr}} \cdot \frac{1}{M} \sum_{j}^{j+M} \left( X_j - \overline{X}_j \right)^2, \tag{4}$$

where $\mathrm{lr}$ refer the size of learning rate. We divide the variance by $\mathrm{lr}$ to normalize and eliminate the effect of $\mathrm{lr}$ scale on CS fluctuations.

We leverage CS-Fluctuation to early stop the fine-tuning process to return a well performing LoRA model. Over the fine-tuning iterations, we empirically observe that the CS-Fluctuations behave like transverse waves, exhibiting multiple peaks and valleys and gradually becoming stable. The first valley of waves often happens at the very beginning of the fine-tuning process at the time when the LoRA model is underfitted. Therefore, we take the second valley as the turning point to signal the early stopping of the fine-tuning process. We then return to users the LoRA checkpoint at the turning-point epoch (one or a few epochs earlier than the current stopping epoch). Kindly note that we apply $M$ size of moving window in Equation 3. Thus, the CS-Fluctuation value of the turning point is calculated based on CS values of previous iterations and those of the subsequent M iterations. The early-stopping LoRA is found in Algorithm 1.

---

**Algorithm 1** Early Stopping based on CS-Fluctuation

1: **Input:** A few reference data, a pre-trained foundation model $\boldsymbol{W}$, Maximum iteration steps J
2: **Output:** A LoRA model at the turning-point epoch ($\boldsymbol{BA}$)
3: **For** j in J:
4:     Compute CS-Fluctuation$_{j-M}$ based on Eq. 4, indexed at $(j - M)$
5:     Compute $\nabla$CS-Fluctuation$_{j-M}$, i.e., the derivative of CS-Fluctuation, for identifying valleys
6:     **If**: $\nabla$CS-Fluctuation experiences a second transition from "-" to "+":          ▷ second valley
7:         Return a LoRA checkpoint at the epoch corresponding to the turning point $(j - M)$
8:         Stop LoRA fine-tuning process

---

## 4 EXPERIMENT

### 4.1 EARLY STOPPING LORA OF LDMS

In this section, we describe the experimental setup and results of using LoRA to fine-tune the LDMs. First, we selected several high-quality LDMs in CivitAI (Civitai, 2022) as base models for LoRA fine-tuning. These models are fine-tuned by Dreambooth on Stable Diffusion V1.5 (SD V1.5), exhibiting superior image generation quality and artistic effects compared with the original version, thus aligning more closely with user needs in real-world applications. To simulate the scenario of users personalizing their private models, we confined the train set to a small scale, i.e., 20-30 images, and did not provide the test set. The image datasets for LDM, supplied by us, included real portraits of an individual, celebrity stills (MovieStillsDB, 2023), landscape of Queenstown in Auckland, and architecture of the Forbidden City. The window size $M$ was configured to the number of steps in an epoch, and the resolution of training images was set to 512*512. We set the Repeat value of 50, indicating that each epoch involves 50 complete traversals of the dataset. The tags were initially

Table 1: Detailed information on image datasets and LoRA fine-tuning hyperparameter settings for LDMs. In real world scenario, users can only provide limited reference data for fine-tuning and test sets are not available since evaluations are subjective.

| MODELS | LORA $r$ | LORA $\alpha$ | LR | BATCH SIZE | REPEAT | EPOCHS |
|---|---|---|---|---|---|---|
| Stable Diffusion V1.5 | 128 | 64 | 1e-5 | 1 | 50 | 10 |

| DATASET | TRAIN SET | TEST SET | | | | |
|---|---|---|---|---|---|---|
| Real portrait | 27 images | N/A | | | | |
| Celebrity | 23 images | N/A | | | | |
| Landscape | 20 images | N/A | | | | |
| Architecture | 22 images | N/A | | | | |

generated by the wd-v1-4-moat-tagger-v2 model (SmilingWolf, 2023) and were subsequently adjusted manually. For more information about datasets and hyperparameter settings, refer to Table 1, and the selected base models in Appendix A.1.

Regarding the hyperparameter of LoRA, LoRA rank, denoted as $r$ in Section 3.1, represents the rank used in the decomposition of the weight matrix. Meanwhile, the $\alpha$ is a scaling constant applied to the output of the low-rank decomposition, i.e., $\boldsymbol{B}\boldsymbol{A}x$ in Equation 1.

We conducted fine-tuning experiments on each image dataset using two different base models. Figure 1a and Figure 3 display the second valley (turning point) identified by Algorithm 1, along with the qualitative experimental results of the trained LoRA models. Due to space constraints, we only present the fine-tuning results of each dataset on one base model, and additional results can be found in Appendix A.1.

The qualitative results reveal that the turning point identified based on CS-Fluctuation can indeed locate well performing LoRA models, thereby avoiding overfitting. Specifically, the LoRA models corresponding to the turning point epoch and the subsequent epoch exhibit superior performance in generating high-quality images based on the prompt while incorporating user-provided references. In contrast, the overfitted LoRA result in generated images that are inconsistent with the prompt, and they degrade the quality of the generated images, leading to blurry and loss of details. For example, in Figure 1a, overfitting starts around Epoch 5, as evidenced by generating an unmentioned "hat" in the prompt. Starting from Epoch 6, the LoRA model begins to ignore the prompts such as "arms crossed", "exposed brickwork", etc., and the background becomes blurred. Similarly, the 'blue hair" prompt is ignored in the first row of Figure 3, the unrealistic snow mountains in the second row, and buildings on the clouds in the third row. For a more detailed view of the image quality, we recommend that readers zoom in on the images or refer to Appendix A.2 for more zoomed-in images.

In summary, the LoRA models that employ early stopping based on CS-Fluctuation can integrate user-provided references into the model without compromising the quality and diversity of the generated images.

## 4.2 EARLY STOPPING LORA OF LLMS

This section illustrates the experimental setup and results regarding the LLMs. We chose the 7B and 13B versions of the LLaMA series, owing to their open-source availability and offering of multi-scale modeling options. The employed text dataset for LLMs is derived from the MMLU dataset (Hendrycks et al., 2020), which contains multiple-choice questions from 57 subjects. We extracted ten subjects and allocated the dev and validation set as the train set to ensure the small size of the training dataset, similar to the LDMs experiments. The window size $M$ was set to 100 steps. Meanwhile, the original test set of the MMLU dataset was utilized to calculate the zero-shot accuracy to quantitatively evaluate the LoRA models at the turning points identified by the CS-Fluctuation. More information about the text dataset and hyperparameter settings can be found in Table 2

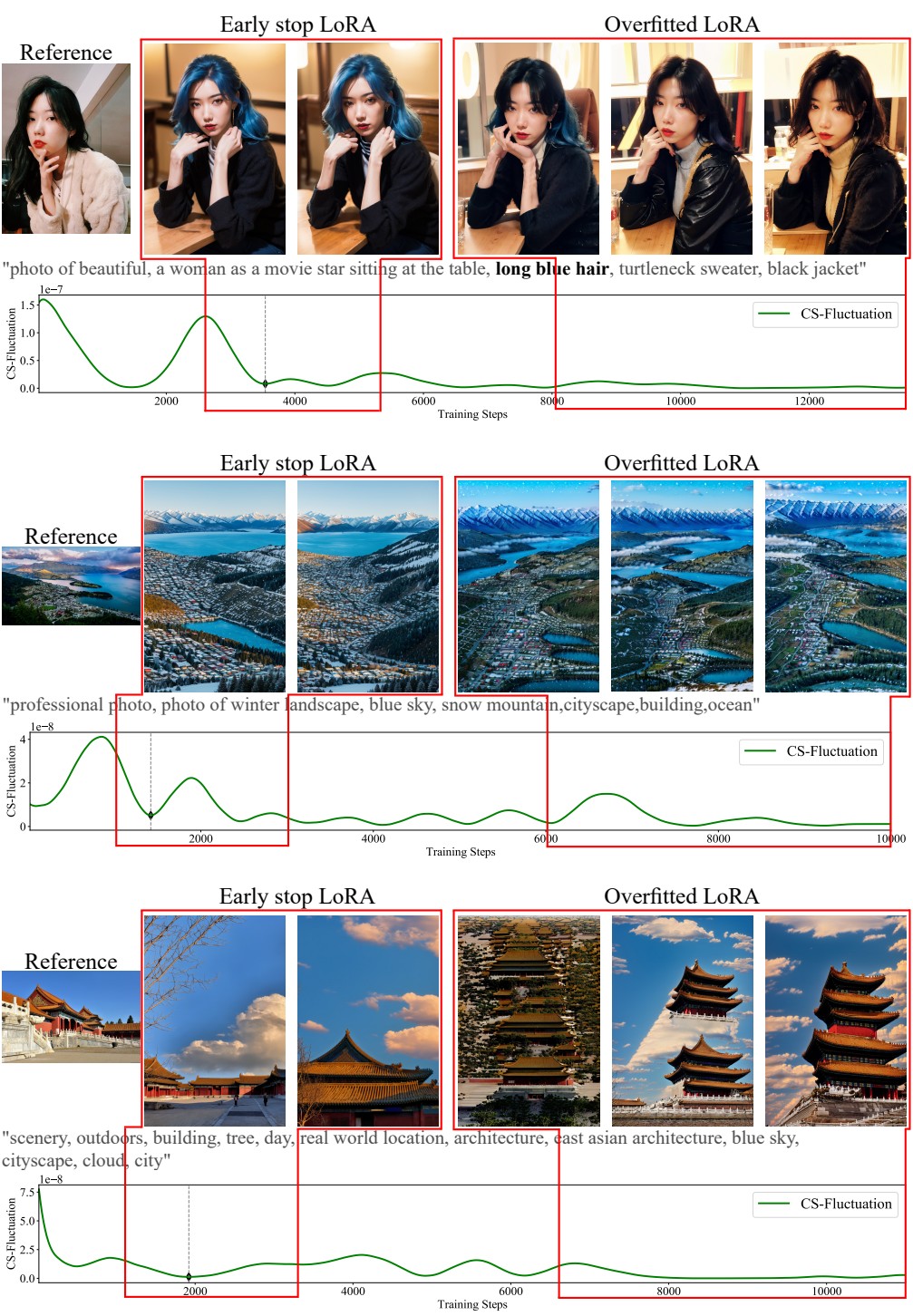

Figure 3: Experimental results for real portrait, landscape, and architecture datasets. Each figure group contains five generated images, the curve of CS-Fluctuation during fine-tuning and the second valley (turning point) marked with the gray dashed line. The first image labeled as "Early stop LoRA" is the generated using the LoRA model at the epoch corresponding to the truning point. The second image is generated using the LoRA model trained for one additional epoch after the turning point epoch. "Overfitted LoRA" refers to the generated results of the LoRA models that continued training after the turning point. For clarity, only part of the prompt is displayed, and non-critical tags such as "masterpiece" and "photorealistic" have been omitted.

Table 2: Detailed information on test datasets and LoRA fine-tuning hyperparameter settings for LLMs

| MODELS | LORA RANK | LORA ALPHA | LR | BATCH SIZE | STEPS |
|---|---|---|---|---|---|
| LLaMA 7B | 64 | 16 | 4e-6 | 16 | 1500 |
| LLaMA 13B | 64 | 16 | 2e-6 | 16 | 1500 |

| DATASETS | SUBJECT | TRAIN SET | Test set |
|---|---|---|---|
| Text dataset | College Physics | 16 | 102 |
| | Machine Learning | 16 | 112 |
| | Clinical Knowledge | 34 | 265 |
| | Business Ethics | 16 | 100 |
| | College Biology | 21 | 144 |
| | Anatomy | 19 | 135 |
| | College Chemistry | 13 | 100 |
| | College Mathematics | 16 | 100 |
| | Computer Security | 16 | 100 |
| | International Law | 18 | 121 |

We conducted LoRA fine-tuning experiments on both versions of LLaMA using train sets from five different subjects for each. The turning points identified by the Algorithm 1 and quantitative results are presented in Figure 4.

The quantitative results validate the effectiveness of CS-Fluctuation in identifying the turning points in the fine-tuning process. Before reaching this point, the accuracy curve fluctuates notably and then stabilizes. For LLMs, the overfitted LoRA model does not significantly affect model performance. However, continuing training after the turning point results in unnecessary computational expenditure, especially since fine-tuning LLMs typically requires more computational resources compared to LDMs.

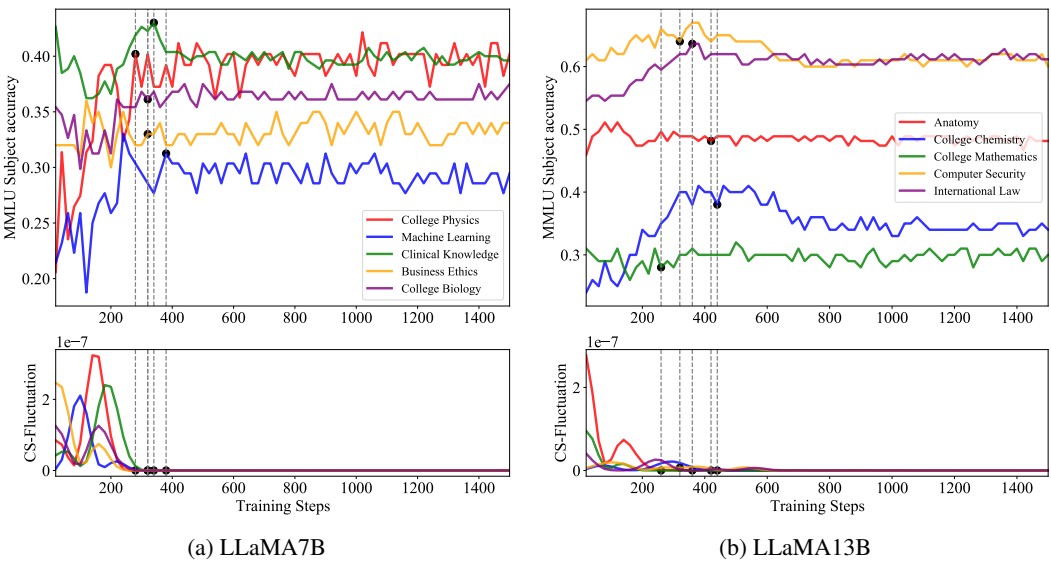

(a) LLaMA7B          (b) LLaMA13B

Figure 4: Quantitative results for LLaMA 7B and 13B, including the zero-shot accuracy calculated using the original test set of the MMLU dataset and the of CS-Fluctuation during the fine-tuning process, along with second valley (turning point) marked with the gray dashed line.

Furthermore, we found that the LoRA model at the turning point is not always the instance with the best performance. Higher accuracy may also exist near the turning points. We compare these in Table 3 with the five-shot baseline accuracy of LLaMA (Touvron et al., 2023a). The accuracy at the

Table 3: Comparison of accuracy during the fine-tuning process. The table displays the zero-shot accuracy (EARLY STOP ACC.) at the second valley (turning point) , the highest accuracy (HIGHEST ACC.) throughout the fine-tuning process, and the five-shot baseline accuracy Touvron et al. (2023a). We find that the best accuracy is usually achieved near the identified turning points.

| MODELS | SUBJECT | EARLY STOP ACC. | HIGHEST ACC. | FIVE-SHOT BASELINE ACC. |
|---|---|---|---|---|
| LLaMA 7B | College Physics | 40.2 | 42.2 | 26.5 |
| | Machine Learning | 31.3 | 33.0 | 23.2 |
| | Clinical Knowledge | 43.2 | 43.2 | 35.1 |
| | Business Ethics | 33.0 | 36.0 | 40.0 |
| | College Biology | 36.1 | 37.5 | 37.5 |
| LLaMA 13B | Anatomy | 48.1 | 51.1 | 45.9 |
| | College Chemistry | 38.0 | 41.0 | 30.0 |
| | College Mathematics | 28.0 | 32.0 | 32.0 |
| | Computer Security | 64.0 | 67.0 | 65.0 |
| | International Law | 63.6 | 63.6 | 62.8 |

identified turning points does not differ significantly from the peak accuracy, the latter frequently materializing near the turning points. Therefore, if the user is not satisfied with the accuracy of the turning points, such points can serve as preliminary stage in a subsequent, more meticulous fine-tuning process.

Moreover, as shown in Table 3, we observe that, in some cases, the zero-shot accuracy of the fine-tuned LoRA models is comparable to the five-shot baseline accuracy, especially in the experiments on LLaMA 13B. This phenomenon may be due to the larger number of parameters than the 13B model, leading to a higher susceptibility to overfitting, particularly when fine-tuning on such a limited dataset.

## 5 LIMITATIONS AND FUTURE WORKS

There are several limitations of this study. Firstly, we have to admit that currently CS-Fluctuation only addresses the overfitting issue of LoRA fine-tuning. As far as we know, other fine-tuning methods, such as Dreambooth, are also prone to overfitting. Secondly, as demonstrated in Table 3, there are some cases that the LoRA has no significantly better performance than five-shot baselines, in which CS-Fluctuation cannot help. In addition, this study does not involve experiments on even larger LLMs (such as LLaMA 33B) due to our computational constraints.

There are several areas deserving further explorations. We plan to apply the early-stopped LoRA in broad applications, such as generating anime characters and landscapes and modifying the image art styles. Besides, we plan to investigate the effectiveness of CS-Fluctuation larger LLMs (e.g., the 65B version of LLaMA and LLaMA2) as well as larger LDMs (e.g., SDXL (Stability-AI, 2023)). Besides LoRA fine-tuning, we plan to develop more robust and generalized methods to mitigate the issue of overfitting in other fine-tuning processes.

## 6 CONCLUSION

In this study, we have introduced a new monitoring metric, CS-Fluctuation, aimed at early stopping the LoRA fine-tuning process of foundation models to avoid overfitting. This approach is particularly valuable in the cases with limited training data or when objective test data is either unavailable or highly subjective. Empirically, we have applied the early-stopped LoRA to both vision models (LDMs) and language models (LLMs), respectively. Our findings corroborate the CS-Fluctuation metric can effectively deliver well-performing, customized vision or language models to users.

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

# A APPENDIX

## A.1 SELECTED BASE MODELS AND ADDITIONAL EXPERIMENTAL RESULTS OF LDMs

For each image dataset, we performed fine-tuning experiments on two base models, as shown in Table 4. All of these base models can both be found and downloaded in CivitAI (Civitai, 2022). The qualitative results for the first base model are presented in Figure 1a and Figure 3, while results for the second base model can be found in Figure 5 and Figure 6.

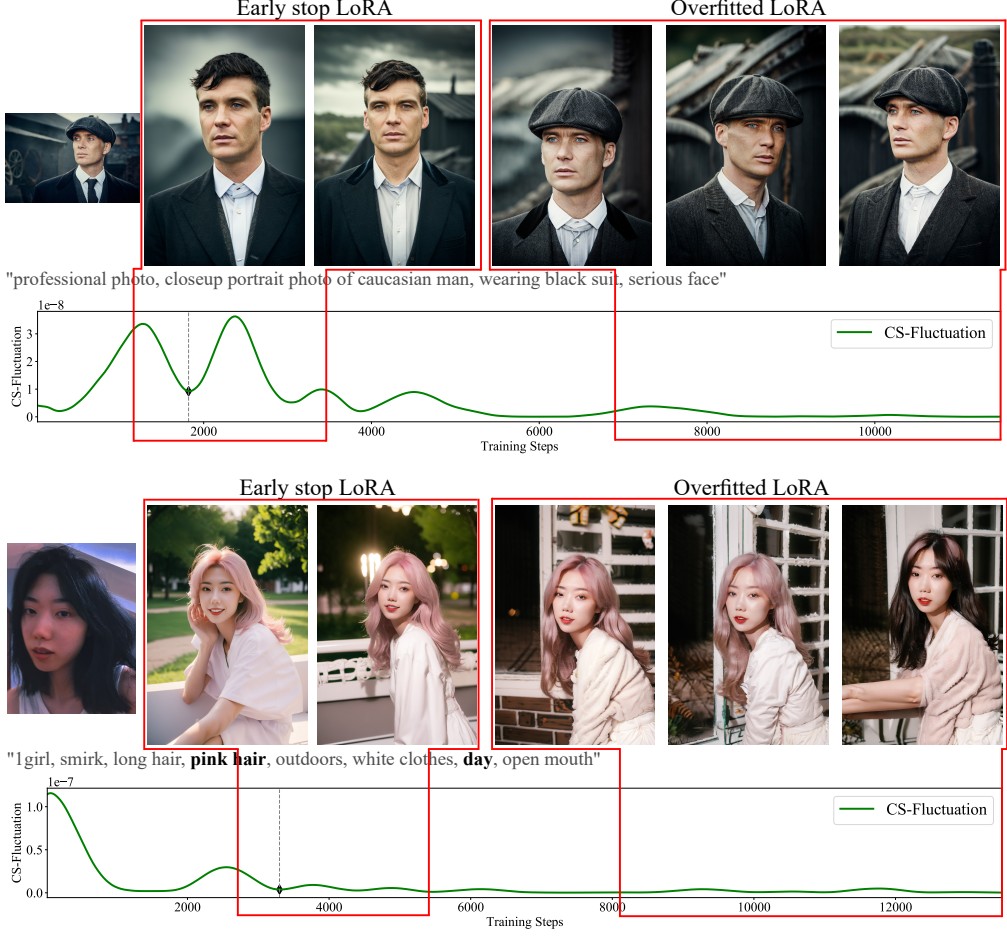

Figure 5: Additional experimental results on celebrity and real portrait datasets. Each figure group contains five generated images, the curve of CS-Fluctuation during fine-tuning and the second valley (turning point) marked with the gray dashed line.

Table 4: Selected Base Models for LDMs Experiemnts

| DATASET | BASE MODEL |
| --- | --- |
| Real portrait | xxmix9realistic_v40 |
| | majicmixRealistic_betterV2V25 |
| Celebrity | icbinpICantBelieveIts_seco |
| | realisticVisionV51_v51VAE |
| Landscape | realisticVisionV51_v51VAE |
| | landscapeRealistic_v20WarmColor |
| Architecture | aargArchitecture_v10 |
| | architecturerealmix_v1repair |

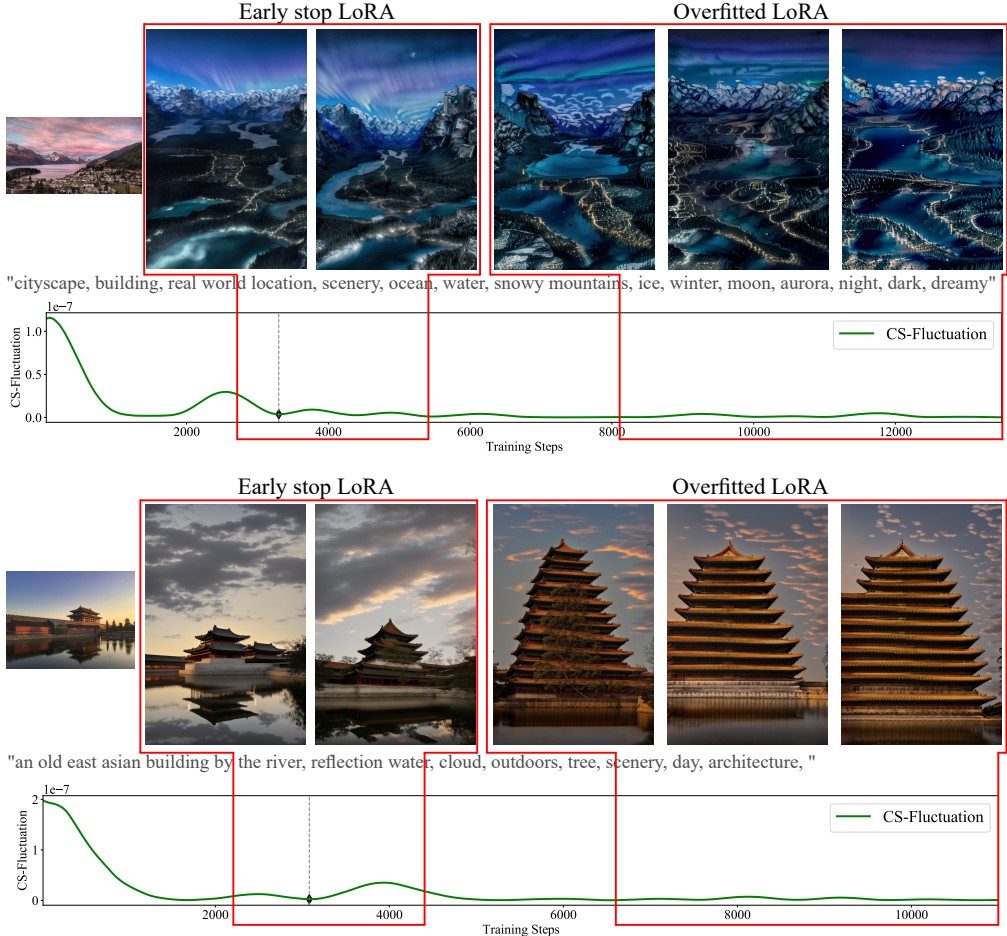

Figure 6: Additional experimental results on landscape and architecture datasets. All images were generated with the same parameters, and non-critical tags have been omitted.

In Figure 5, we observe that the overfitted LoRA model in the first group of images generates a character wearing a hat, even though "hat" is not provided in the prompt. Furthermore, the overfitted LoRA generates images that resemble the user-provided references. Besides, in the second group, the model ignores the prompt "day" and "pink hair", and significantly degrades the quality and diversity of the generated images.

In Figure 6, the second base model of landscape tends to generate dreamy style images rather than pursuing image realism. However, overfitting still impacts image quality, i.e. the auroras and snow mountains are manifested in anomalous formations. Also for the architecture, weird and unusual buildings are generated.

## A.2    ADDITIONAL QUALITATIVE RESULTS

In Figure 7, 8, 9, and 10 we present more qualitative results for LDMs, including the generation results of the well performing LoRA model (Early stop LoRA) at turning point and the overfitted results. For clarity, we have zoomed in on the images and non-critical tags have been omitted.

Early stop LoRA                          Overfitted LoRA

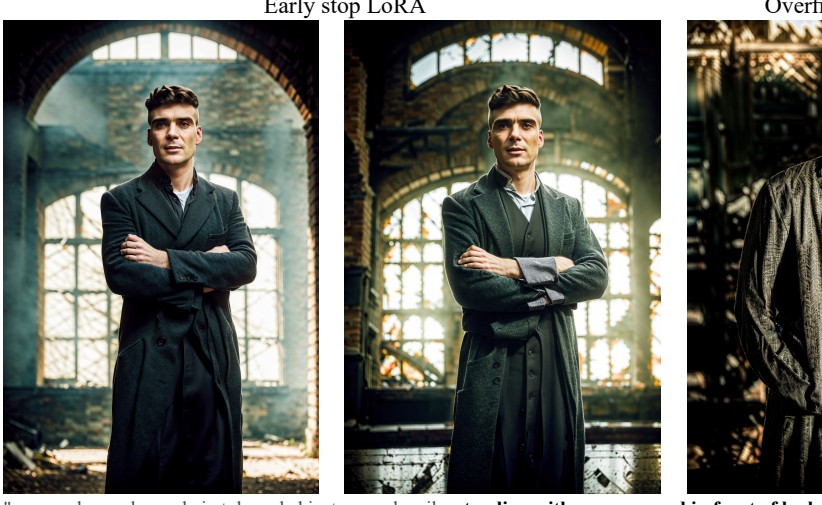

"a man, shaggy layers hairstyle and chinstrap, sad smile, **standing with arms crossed in front of body** in WWII,Bombed-Out Building Interior, **Exposed brickwork**, **rubble**, **damaged furniture**, evidence of previous life"

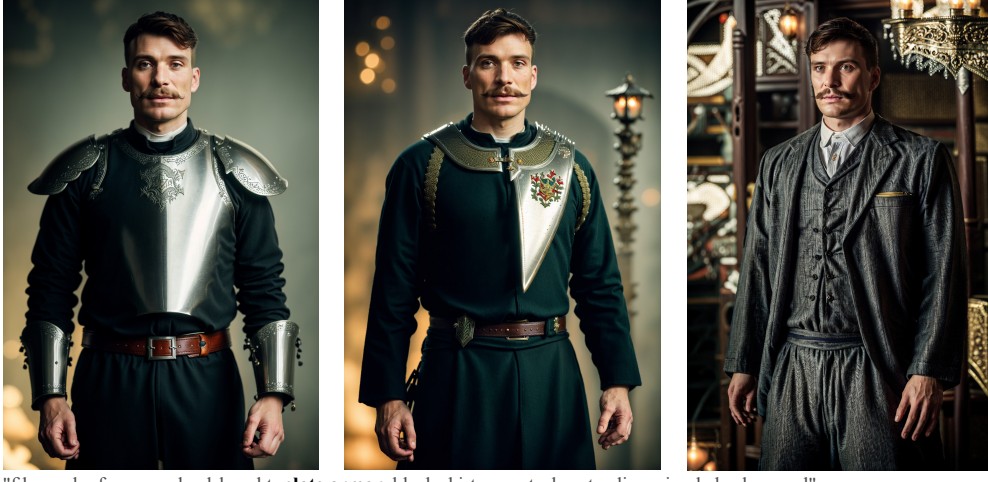

"film grab of a young landsknecht, **plate armor**, black shirt, moustache, standing, simple background"

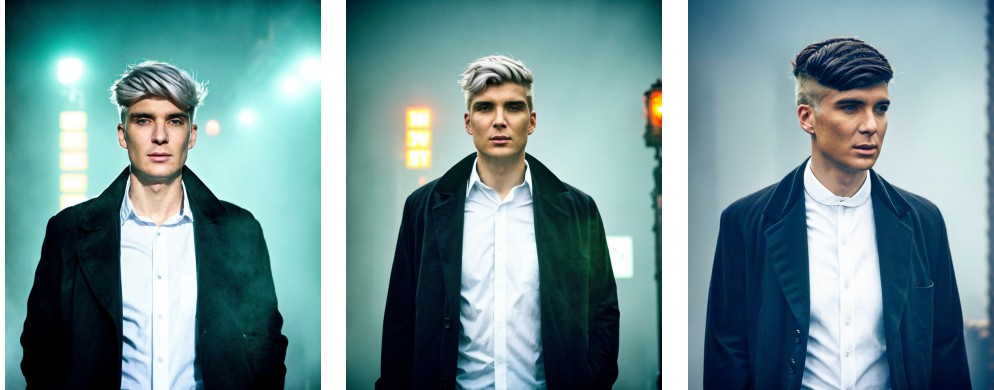

"city street, fog, closeup portrait photo of young man wearing white shirt and black jacket, **white hair**"

Figure 7: Additional qualitative results on celebrity dataset

Early stop LoRA                          Overfitted LoRA

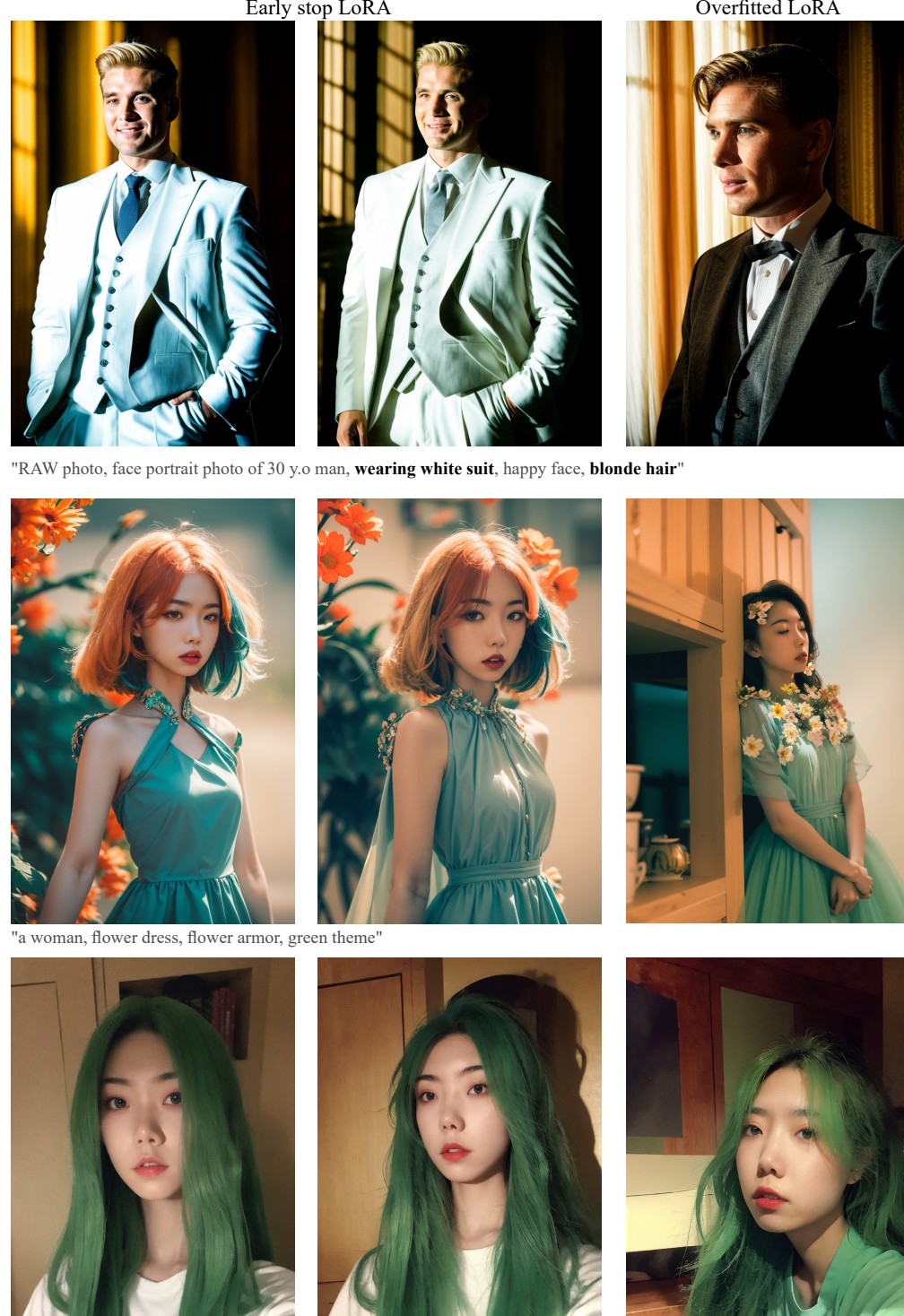

"RAW photo, face portrait photo of 30 y.o man, **wearing white suit**, happy face, **blonde hair**"

"a woman, flower dress, flower armor, green theme"

"20 yo woman, long straight hair, green hair, **white shirt**"

Figure 8: Additional qualitative results on celebrity and real portrait dataset

Early stop LoRA                                                    Overfitted LoRA

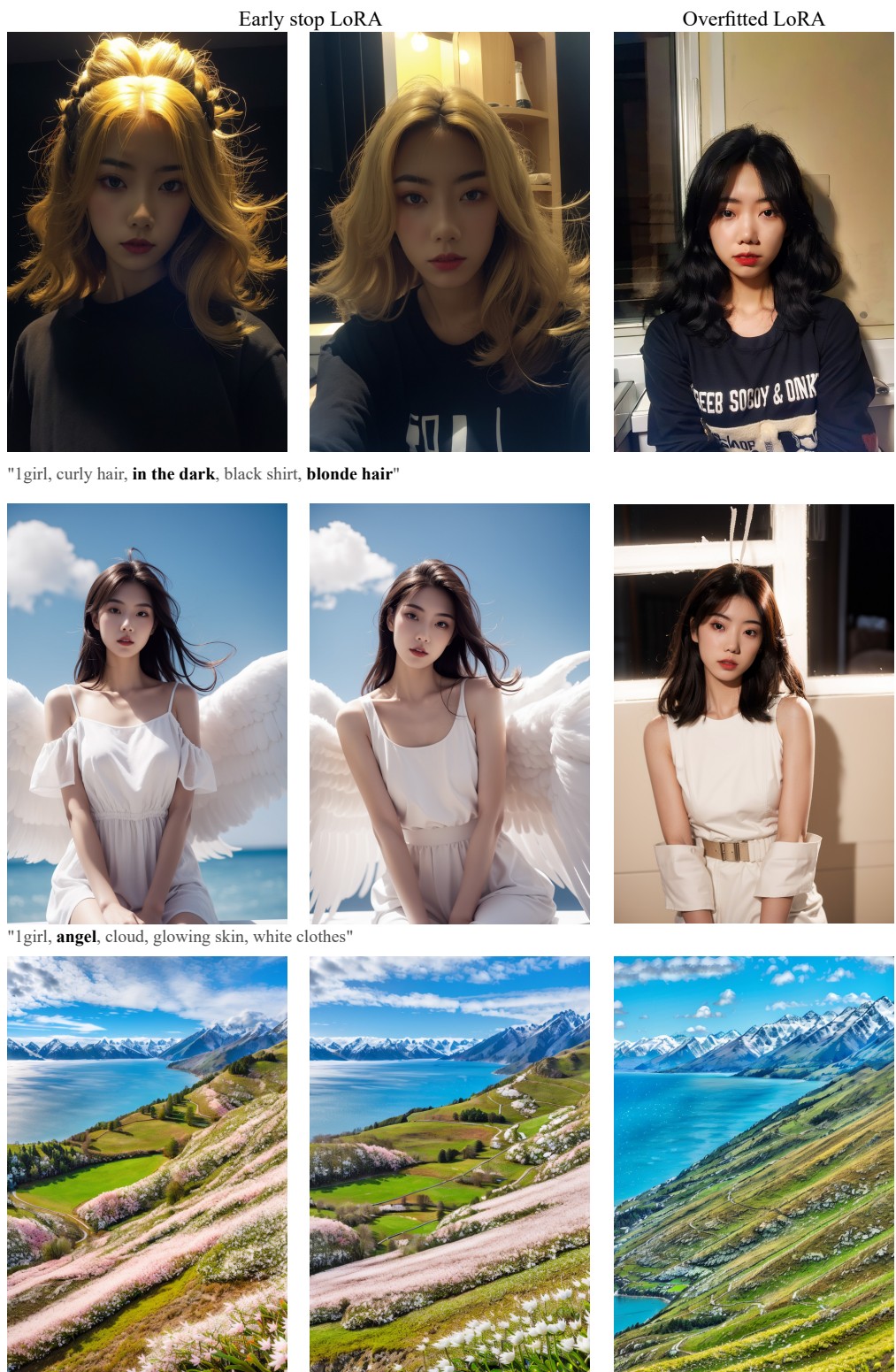

"1girl, curly hair, **in the dark**, black shirt, **blonde hair**"

"1girl, **angel**, cloud, glowing skin, white clothes"

"photo of spring landscape, blue sky,ocean,snow mountain,cloudy sky, **flowers**"

Figure 9: Additional qualitative results on real portrait and landscape datasets

Early stop LoRA                                    Overfitted LoRA

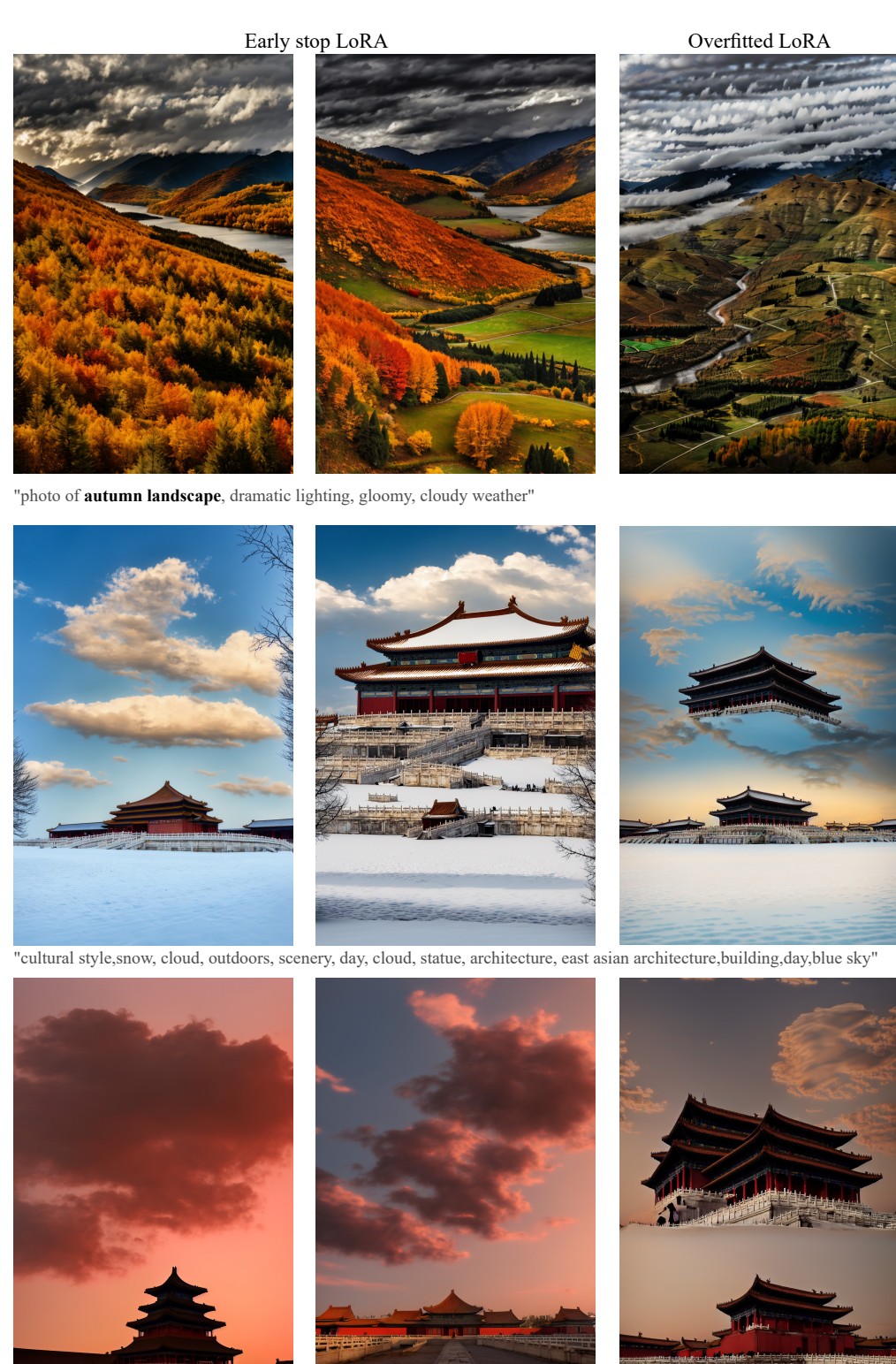

"photo of **autumn landscape**, dramatic lighting, gloomy, cloudy weather"

"cultural style,snow, cloud, outdoors, scenery, day, cloud, statue, architecture, east asian architecture,building,day,blue sky"

"scenery, outdoors, building, day, real world location, architecture, east asian architecture, dusk, cloud, cityscape,city"

Figure 10: Additional qualitative results on landscape and architecture datasets

