# OpenReview forum: "Towards Personalized AI: Early-stopping Low-Rank Adaptation of Foundation Models"
_ICLR.cc/2024/Conference — Submitted to ICLR 2024_

### Official Review · Reviewer_9V6J · 2023-10-30

**Soundness:** 1 poor
**Presentation:** 2 fair
**Contribution:** 3 good
**Rating:** 3
**Confidence:** 4

**Summary:**

The paper proposes CS-Fluctuation, a monitoring metric for early stopping the fine-tuning of foundation models. This is a validation-independent criterion that can be useful in settings where there is a deficiency of validation data. The method has been tested on LDMs and LLMs.

**Strengths:**

- The paper is well organized and easy to follow.
- The motivation is straightforward and important
- Many qualitative examples are provided

**Weaknesses:**

- No theoretical background or explanation on why CS-Fluctuation is a good indicator of overfitting. Why does this work?
- As CS-Fluctuation is a strictly empirical criterion, more quantitative experiments and analyses is needed to support the validity of this early-stopping method. Also, what happens if you vary N in the N-th valley early stopping?
- Qualitative examples are not enough to demonstrate whether the model has been overfitted or not. Many of the samples that the authors have labeled “Overfitted LoRA” does not seem to be particularly overfitted (e.g. Figure 3 top / middle’s center image, Figure 5 top / bottom 3rd, 4th image etc). Quantitative comparison on LDMs is necessary to make the authors’ claim convincing.

**Questions:**

- Does the Five-shot baseline in Table 3 refer to the case of using 5-shot samples as the validation set? If not, I would want to see a comparison of CS-Fluctuation based early stopping and the standard validation set based early stopping

---

> ### Author Response · Authors · 2023-11-14
>
> Dear Reviewer,
>
> Thank you so much for your valuable comments and suggestions. We would like to respond in detail regarding the issues you raised:
>
> 1.Theoretical Foundation of the Method:
> We recognize that the current approach may lack a complete theoretical framework. However, through a series of experiments, we have found that using CS-Fluctuation as a metric is significantly effective. The cosine similarity effectively measures the change in direction between model weight vectors, thus reflecting the learning state of the network. Experimental results show that this method is effective in preventing overfitting during fine-tuning, and the role of empirical evidence cannot be ignored.
>
> 2.Quantitative Analysis:
> The field of personalized AI in fine-tuning the foundation models and preventing overfitting is still in a developmental stage. LDMs models currently have no effective metrics to check if the model is overfitting, we can only use the generated results to make a manual judgment.
>
> Besides, Our approach is based on the training step deciding to early stop, and returning the LoRA model for the epoch of the corresponding training step, and using it to generate qualitative results (generating images). Therefore, the "overfitted LoRA" results we show are exactly what you would expect after several valleys. As the training progresses, the overfitting of the model will gradually increase, leading to further degradation of the quality of the generated images.
>
> 3.Explanation on Qualitative Results:
> Our goal is to obtain personalized text-to-image models with high generation quality, which should efficiently incorporate references (e.g., faces) provided by individuals, while maintaining their original text-guided generation capabilities. Overfitted LoRA models typically exhibit lower image quality, such as low pixel, blurred, or unrealistic features (as shown in Figure 1 and Figure 3). In addition, overfitting also affects the text-based generation ability of the model, as shown by "arms crossed" in Figure 1 and "long blue hair" in the first line of Figure 3.
>
> Thank you again for your review and valuable comments. We look forward to further guidance and suggestions.

---

### Official Review · Reviewer_fqrQ · 2023-10-30

**Soundness:** 2 fair
**Presentation:** 2 fair
**Contribution:** 2 fair
**Rating:** 3
**Confidence:** 3

**Summary:**

This paper aims to address the overfitting issue that arises when fine-tuning a pre-trained foundation model using Low-Rank Adaption (LoRA). Particularly, the authors proposed a new metric called CS-Fluctuation, based on the cosine similarity between the fixed model weight and the added trainable weight using personal data.

**Strengths:**

1. The proposed method CS-Fluctuation is very simple and kind of reasonable from the case study in this paper.
1. The proposed method are demonstrated in various benchmark foundation models.

**Weaknesses:**

1. The proposed method lacks intuitive understanding and theoretical guarantee. It is hard to fully understanding why the proposed method is reasonable, especially why the metric is based on the cosine similarity between the model weights?
1. The scale of the metric ($\approx1e-7$) is too small and changes from data to data. Some normalization is needed for the metric.
1. The experiment demonstration is monotonous, only CS-Fluctuation vs. training steps is showcased, more aspects about the proposed method should be presented to justify the claims.
1. No qualitative comparisons. It hard to judge the superiority of the proposed method.

**Questions:**

1. In figure 1, what is the connection between Epoch and training steps? It seems the first two epoch is sufficient for model fine-tuning from the figure?
1. From the figure 3, it seems there is no clear signal which training steps is better. Why not early stop the method at the first $K$ epoch? set $K=2$ according figure 1.

---

> ### Author Response · Authors · 2023-11-14
>
> Dear Reviewer,
>
> Thank you so much for your valuable comments and suggestions. In response to your concerns, we provide the following explanations and answers:
>
> Regarding the Weakness:
>
> 1.Theoretical Foundation of the Method:
> Although our method may lack the traditional full theoretical framework, it has demonstrated significant results in a series of experiments. We chose cosine similarity as the metric mainly because it is a straightforward and effective measure of the directional change between model weigh vectors, thus reflecting the learning state of the network. Experimental results show that this approach is effective in preventing overfitting during fine-tuning, and the role of empirical evidence cannot be ignored.
>
> 2.Normalization of CS-Fluctuation:
> As shown in Equation 4, we have already normalized the learning rate to eliminate the effect of learning rate on CS-Fluctuation. We agree that further normalization of CS-Fluctuation is necessary to ensure its validity in different settings.
>
> 3.Comparative Analysis:
> The field of personalized AI in fine-tuning the foundation models and preventing overfitting is still in a developmental stage, and we do not include comparative analysis with other approaches in our paper due to the lack of directly comparable existing work.
>
> Regarding the Questions:
>
> 1.Relationship Between Epoch and Training Steps:
> Epoch and training steps actually correspond to each other and represent different ways of controlling training process. The formula is: Training steps = Repeat * Train set size * Epochs / Batch Size. In Figure 1, the better-performing LoRA models are the results of Epoch 2 and Epoch 3. This conclusion can also be reached and validated through manual judgment of the results generated by the LoRA model.
>
> When dealing with a large number of personalized modeling tasks, the absence of a metric like CS-Fluctuation would result in having to manually selection for each tasks, leading to significant time consumption.
>
> 2.Strategy of Early Stopping:
> Our approach is based on training steps to decide on early stopping and returns the LoRA model of the epoch in which the selected training step is located. We do not use the model at the exact training step moment because it may be overfitted to the current batch of data.
>
> Regarding why we do not directly early stop at a fixed epoch (e.g., K=2), the reason is that the model training results are affected by Repeat factor. We set all repeats to 50 (indicating the number of times each image is used in each epoch). If the repeat value is set higher or lower, a fixed K value would no longer be applicable.
>
> Thank you again for your review and valuable comments. We look forward to further guidance and suggestions.

---

> > ### Comment · Reviewer_fqrQ · 2023-11-23
> > **Thanks for your efforts**
> >
> > Thanks for the authors' rebuttal. I have no concerns about the technical aspects of the proposed method. However, I still have concerns about the presentation of the current manuscript, which lacks intuitive understanding and qualitative comparisons, making it hard to judge the superiority of the proposed method, as empirical results are just one aspect of the method. Therefore, I will not change my score, but I sincerely hope my comment can be helpful to make the current version better.

---

### Official Review · Reviewer_rWot · 2023-10-31

**Soundness:** 3 good
**Presentation:** 3 good
**Contribution:** 3 good
**Rating:** 6
**Confidence:** 4

**Summary:**

The paper introduces CS-Fluctuation, a novel metric for identifying the optimal point during personalized foundation model fine-tuning. This metric enables early stopping to prevent overfitting, especially when applying Low-Rank Adaptation to small datasets. Experiment results on vision and language models confirm CS-Fluctuation's effectiveness in generating high-quality images and accurate text predictions. This metric has the potential to assist non-AI experts in avoiding overfitting and reducing computational costs.

**Strengths:**

1. Originality: The paper introduces a novel monitoring metric and a fresh approach to pinpoint the turning point in the fine-tuning process, effectively preventing overfitting and reducing unnecessary computational expenses.

2. Quality: The paper provides a detailed algorithm description and comprehensive experimental results, thoroughly validating the effectiveness of the proposed metric and method, spanning both vision and language foundation models.

3. Clarity: The paper is well-structured and maintains clarity in the calculation of CS-Fluctuation. The diagrams illustrating the experiment setup and figures depicting the experimental results further enhance the paper's clarity.

4. Significance: This metric and approach hold particular significance, especially when dealing with limited training data or situations where objective test data is either unavailable or subject to high subjectivity.

**Weaknesses:**

I overall appreciate the novel idea and significant contribution of this paper, while still having some concerns in the implementation and experimental settings.

1. The proposed metric is effective exclusively for LoRA and doesn't extend to other fine-tuning methods.

2. Excessive moving window average operations may overly smooth results, potentially missing subtle yet important changes or trends.

3. Selecting the second through as the turning point appears somewhat speculative and lacks sufficient mathematical and theoretical explanations or proofs. Its applicability in all cases remains uncertain, and the paper does not offer examples of failure cases.

4. The datasets for experiments on diffusion models are supplied by the authors. Using widely recognized datasets might enhance the reliability and persuasiveness of the results.

5. Experiments with language models exclusively involve LLAMA models, with no inclusion of larger or different types of language models. Additionally, only a portion of the MMLU dataset is used.

6. The paper lacks specific experiment details, such as image tags and the choice of optimizer.

7. In experiments with language models, some LoRA models fail to outperform the original, unfine-tuned models, calling into question the viability of fine-tuning large language models with such limited datasets.

**Questions:**

Refer to Weaknesses.

---

> ### Author Response · Authors · 2023-11-14
>
> Dear Reviewer,
>
> Thank you very much for your valuable comments and suggestions. Regarding the issues you mentioned, we would like to respond as follows:
>
> 1.The concerns you point our are indeed important, and our current approach is mainly applicable to LoRA, but we view it as a significant starting point and believe that the core concepts can be applied to other fine-tuning approaches, which is the focus of our future work.
>
> 2.We have conducted many experiments on this point, and due to the use of limited dataset to train such a larger network, the training process can be unstable with many minor fluctuations. The moving window average techniques aims to balance the stability and sensitivity of the results to avoid misjudgments due to minor fluctuations. Our experimental results have proven this method to be effective in practice.
>
> 3.Regarding the choice of the second valley, we recognize that it may seem subjective. However, this choice is based on extensive experiments and analyses. We have found that he first valley typically occurs in the early stage of fine-tuning, when the model has not yet fully adapted to the data. The second valley, on the other hand, provided a more reliable indicator that the model is beginning to overfit.
>
> 4.Our choice to use specific datasets for experiments is intended to better simulate real-world applications of personalized AI. This design allows anyone to replicate our experiment results using their own photos to personalize the foundation models.
>
> 5.The LLAMA models are selected because of their open-source nature and the availability of different size versions. Meanwhile, your point is reasonable, and in future work we intend to include more experiments on downstream tasks of LLMs.
>
> 6.The code provides all the experimental details, which are not all shown in the paper due to space issues.
>
> 7.Regarding the issue of some LoRA models not outperforming the original, un-fine-tuned models, we have discussed this in our paper. This may be due to larger models being more prone to overfitting, especially when fine-tuned on such limited datasets.
>
> Thank you again for your review and valuable comments. We look forward to further guidance and suggestions.

---

### Official Review · Reviewer_ndtG · 2023-10-31

**Soundness:** 2 fair
**Presentation:** 3 good
**Contribution:** 2 fair
**Rating:** 5
**Confidence:** 3

**Summary:**

Foundation models have demonstrated impressive performance in a wide range of practical applications after being trained on extensive datasets. Fine-tuning these pre-trained models is a cost-effective way to adapt them to specific, smaller datasets, but non-experts often struggle with hyperparameter settings and risk overfitting without realizing it. To address this challenge, the paper introduces a novel monitoring metric called CS-Fluctuation, which aids in early stopping during the fine-tuning process. This approach combines Low-Rank Adaptation (LoRA) to fit personalized data while continuously monitoring the cosine similarity of parameter changes between the LoRA branch and its corresponding layer. When these changes stabilize, it signals the onset of overfitting, which becomes more pronounced as fine-tuning progresses. Empirical experiments with various types of personalized data on both vision and language foundation models confirm the effectiveness of CS-Fluctuation in early stopping LoRA fine-tuning.

**Strengths:**

Importance of the Research Problem: The paper addresses a significant issue in the field of AI, namely the challenge of personalizing foundation models. Personalized AI has become increasingly relevant in various applications, making the problem studied in the paper highly important.

Interesting Idea: The concept of early-stopping low-rank adaptation of foundation models is intriguing. It introduces a novel approach to the problem, which could have practical implications in improving the efficiency of personalized AI systems.

Experimental Validation on Multiple Datasets: The authors have conducted experiments on multiple datasets, demonstrating a comprehensive evaluation of their proposed method. This multi-dataset validation enhances the credibility and applicability of their findings.

**Weaknesses:**

Limited Contribution: While the problem studied is important, the paper may be lacking in terms of innovation. The proposed method, early-stopping low-rank adaptation, may need some inspiration from a theoretical perspective, highlighting how it offers a unique and innovative solution.

Unclear Generalization of Metrics: The paper introduces certain metrics, but it's not clear how these metrics can be generalized to other AI applications or datasets. A more thorough discussion of the potential transferability and generalization of the proposed metrics would enhance the paper's impact.

Lack of Real-World Application Discussion: The paper could benefit from a deeper discussion of real-world applications and scenarios where the proposed method might be particularly advantageous. Providing practical use cases and illustrating how the method could address real AI problems would add value to the research. Also, it is interesting to explore the performance of CS-Fluctuation for other tuning techniques.

In conclusion, the paper addresses an important issue in personalized AI and presents an interesting idea with experimental validation on multiple datasets. However, it may need to emphasize the innovation of its proposed method, clarify the generalization of introduced metrics, and provide more context about real-world applications to strengthen its contributions.

**Questions:**

Limited Contribution: While the problem studied is important, the paper may be lacking in terms of innovation. The proposed method, early-stopping low-rank adaptation, may need some inspiration from a theoretical perspective, highlighting how it offers a unique and innovative solution.

Unclear Generalization of Metrics: The paper introduces certain metrics, but it's not clear how these metrics can be generalized to other AI applications or datasets. A more thorough discussion of the potential transferability and generalization of the proposed metrics would enhance the paper's impact. Also, it is interesting to explore the performance of CS-Fluctuation for other tuning techniques.

Lack of Real-World Application Discussion: The paper could benefit from a deeper discussion of real-world applications and scenarios where the proposed method might be particularly advantageous. Providing practical use cases and illustrating how the method could address real AI problems would add value to the research.

---

> ### Author Response · Authors · 2023-11-14
>
> Dear Reviewer,
>
> Thank you very much for your valuable comments and suggestions. We have carefully considered the issues you raised and will respond to each of them in turn.
>
> 1.Innovation and Contribution: Although our research does employ the pre-existing LoRA method, our innovation extends beyond this. The core innovation of our work is the introduction of a new metric and a unique approach focused on detecting overfitting and early stopping during the fine-tuning process of foundation models. This is a common but unsolved challenge in current research. We not only identified this challenge, but also developed an innovation approach to monitor and avoid the overfitting issues, which is quite rare in existing studies.
>
> 2.Generalization of CS-Fluctuation: Despite the fact that our experiments are based on some specific datasets, the design concept and methodology behind them have universally applicability. Our experiments all simulate individuals personalizing the foundation model using the limited data they provide. Based on this design, each individual can reproduce the results of the experiments with their own photos.
>
> 3.Real-world Applications: Our work has mentioned two real-world application scenarios, such as individuals who want text-to-image models to generate personalized, high-quality face images, or specialized LLMs models for specific domains. These applications are currently pretty common in the field of personalized AI, and their corresponding experimental results also demonstrate the value and applicability of our approach in solving real-world problems.
>
> Once again, thank you for your review and valuable feedback. We look forward to further suggestions from you.

---

> > ### Comment · Reviewer_ndtG · 2023-11-22
> >
> > Thanks for the detailed response. I think the proposed CS-Fluctuation needs more theoretical guarantees or experimental validation on more real-world applications. It is not convincing that this approach is general. Thus I decide to maintain my score (5).

---

### Meta-Review · Area_Chair_EEVM · 2023-12-06

**Metareview:**

The paper under review introduces CS-Fluctuation, a novel metric aimed at optimizing the fine-tuning process of foundation models through early stopping, specifically in the context of Low-Rank Adaptation (LoRA) for personalized data. While the paper tackles a pertinent issue in AI with a novel approach and demonstrates it through comprehensive experiments, it falls short in theoretical depth, generalizability, and practical applicability.

The reviewers' concerns about its limited theoretical contribution, unclear generalization of metrics, and lack of robust real-world application discussion, alongside questions on experimental design and results, suggest that the paper may not meet the standards for acceptance in its current form. Therefore, the recommendation is for rejection, with encouragement for the authors to address these gaps in future revisions.

**Justification For Why Not Higher Score:**

- The reviewers uniformly express concerns regarding the theoretical foundation and innovation of the method.
- There is a lack of clarity on the metric's generalizability, and concerns about its exclusive applicability to LoRA.
- Reviewers raise questions about the choice of datasets, the scope of experiments, and the interpretation of results

**Justification For Why Not Lower Score:**

N/A

---

### Decision · Program_Chairs · 2024-01-16

Reject